# Microglia Diversity in Healthy and Diseased Brain: Insights from Single-Cell Omics

**DOI:** 10.3390/ijms22063027

**Published:** 2021-03-16

**Authors:** Natalia Ochocka, Bozena Kaminska

**Affiliations:** Laboratory of Molecular Neurobiology, Nencki Institute of Experimental Biology of the Polish Academy of Sciences, 02-093 Warsaw, Poland; n.ochocka@nencki.edu.pl

**Keywords:** microglia heterogeneity, disease-associated microglia, malignant gliomas, glioma associated microglia/macrophages, single-cell RNA sequencing, mass cytometry

## Abstract

Microglia are the resident immune cells of the central nervous system (CNS) that have distinct ontogeny from other tissue macrophages and play a pivotal role in health and disease. Microglia rapidly react to the changes in their microenvironment. This plasticity is attributed to the ability of microglia to adapt a context-specific phenotype. Numerous gene expression profiling studies of immunosorted CNS immune cells did not permit a clear dissection of their phenotypes, particularly in diseases when peripheral cells of the immune system come to play. Only recent advances in single-cell technologies allowed studying microglia at high resolution and revealed a spectrum of discrete states both under homeostatic and pathological conditions. Single-cell technologies such as single-cell RNA sequencing (scRNA-seq) and mass cytometry (Cytometry by Time-Of-Flight, CyTOF) enabled determining entire transcriptomes or the simultaneous quantification of >30 cellular parameters of thousands of individual cells. Single-cell omics studies demonstrated the unforeseen heterogeneity of microglia and immune infiltrates in brain pathologies: neurodegenerative disorders, stroke, depression, and brain tumors. We summarize the findings from those studies and the current state of knowledge of functional diversity of microglia under physiological and pathological conditions. A precise definition of microglia functions and phenotypes may be essential to design future immune-modulating therapies.

## 1. Brief Introduction to Microglia Ontogeny, Turnover and Functions

### 1.1. Microglia Ontogeny, Distribution and Turnover

Microglia—the innate immune cells of the central nervous system (CNS)—belong to CNS macrophages that encompass microglia and CNS border-associated macrophages (BAMs). Early studies using immunohistochemistry demonstrated that microglia represent about 10% of the adult brain cell population and are present in all main CNS structures, although these cells are not uniformly distributed [1]. In initial studies, staining for microglia markers F4/80 and receptors of FcIgG1/2b (Fc, fragment crystallizable), as well as type-three complement (Mac-1) showed two types of positive cells in the adult mouse brain: microglia and cells associated with the choroid plexus, ventricles, and leptomeninges [2]. There is regional diversity in the abundance of microglia; more positive cells are found in the gray matter than the white matter. Microglia are numerous in the hippocampus, olfactory telencephalon, basal ganglia, and substantia nigra. The less densely populated areas include fiber tracts, cerebellum, and the brainstem, while the cerebral cortex, thalamus, and hypothalamus have average cell densities. The proportion of microglia varies from 5% in the cortex and corpus callosum to 12% in the substantia nigra.

Regional heterogeneity of microglia has been detected in the adult human brain, with lower densities in the gray matter compared with the white matter, and substantially lower densities in the cerebellar cortex compared to the substantia nigra [3,4,5]. Common immunological markers include: CD (cluster of differentiation) 45, CD68, HLA-DR (a MHC class II cell surface receptor), IBA-1 (Ionized calcium-binding adaptor molecule 1). Those microglia slowly renew at a median rate of 28% per year, and only 2% of microglia are thought to be proliferating at a given time [6,7]. Most studies in humans assessed microglial populations in embryonic/fetal tissues with immunological (CD45, MHCII, CD68, IBA-1) or histochemical markers (RCA-1) that do not distinguish between microglia and non-parenchymal macrophages.

The issue of microglia maintenance in adulthood has been addressed in studies using bone marrow (BM) cell transplantation and parabiosis. The BM transplantation experiments in rats showed that solely perivascular macrophages are replaced and not the ramified microglia in the brain parenchyma [8]. In female human patients who underwent sex mismatched BM transplantation, only donor-derived perivascular macrophages were detected [9]. In the early cell transfer experiments with transplantation of the BM hematopoietic cells expressing green fluorescent protein (GFP), some GFP-expressing parenchymal microglia were found in the cerebellum, striatum, and hippocampus, suggesting a partial substitution by peripheral cells [10,11]. However, in later studies, in which an animal head was shielded to protect from irradiation, any significant infiltration of BM-derived cells into the brain was not observed, suggesting that the spotted substitution was due to radiation [12]. The experiments on parabiotic mice, in which the turnover of hematopoietic cells for prolonged periods can be studied, demonstrated a lack of microglia progenitor recruitment from the circulation in denervation or CNS neurodegenerative disease. Further studies combining parabiosis and myeloablation showed that recruited monocytes do not persist in the CNS [13]. The expression of a progenitor marker, nestin, is corroborating evidence that microglial progenitors were not hematopoietic cells. All those findings pointed to the persistence of microglia in CNS and a lack of the significant contribution from BM-derived monocytes [14].

In vivo lineage tracing studies, using the fractalkine receptor encoding gene Cx3cr1gfp/+ knock-in mice, established that microglia have a different ontogeny from mononuclear phagocytes and colonize CNS early in the development. Several studies demonstrated that microglia are derived from primitive hematopoietic progenitors (c-Kit^lo^CD41^lo^ progenitors) that originate around embryonic day 7.25 (E7.25) in the yolk sac [15,16,17]. A recent study found two populations: non-Hoxb8 microglia and Hoxb8 microglia, with the latter derived from the second wave of yolk sack hematopoiesis infiltrating the murine brain around E12.5 [18]. In the zebrafish embryo, microglia derive from c-myb-independent erythro-myeloid progenitors but are replaced by c-myb-dependent hematopoietic stem cells (HSC) after birth [19]. These two studies suggest a second wave of proliferation and CNS colonization by microglial progenitors in the early CNS development. All the data point to the origin of microglia from the yolk sack hematopoiesis, early CNS colonization, and the lack of significant input from HSCs in adulthood. Similarly to microglia, BAMs are derived from hematopoietic precursors during embryonic development and establish stable populations, with the exception of choroid plexus macrophages, which have a shorter life span and are replenished from blood-borne monocytes [20].

Due to different ontogeny and location, microglia acquire a different gene expression signature than BAMs and peripheral macrophages [21]. In adulthood, microglia are dependent on constant stimulation of colony-stimulating factor-1 (Csf1) receptors. In both mice and humans, interleukin-34 (IL-34), an alternative ligand for Csf-1 receptor produced by neurons in the brain, is essential for microglia maintenance [22,23].

The initial colonization of the brain by microglia corresponds to development of its vascularization, although microglia may enter the brain via brain ventricles or across meninges [24]. In the human brain, microglia colonization coincides with the vascularization, radial glia formation, neuronal migration, and myelination [25]. The density of microglia in the developing CNS is two times higher than in the adult CNS, and the time-regulated decline in microglia number involves increased apoptosis and reduced proliferation [26]. Microglia acquire a definitive local density soon after birth, after a wave of microglial proliferation within the first 2 postnatal weeks followed by a decline by 50% between the third and sixth postnatal weeks [26].

Studies of microglia in human brain sections from embryos and fetuses from early gestational weeks (gw) showed that microglial cells penetrate into and spread throughout the cortical gray and white matter with the specific spatiotemporal pattern during the first 2 trimesters of gestation. Amoeboid microglia positive for IBA1 (ionized calcium-binding adapter molecule 1), CD68, and CD45 were present in the forebrain starting from 4.5 gw. They penetrate the telencephalon and diencephalon via the meninges, choroid plexus, and ventricular zone. A second wave of microglial cells penetrates the brain via the vascular route at about 12-13 gw and remains confined to the white matter [27,28].

There are some indications that microglia turnover in mice is slowed down with aging. Studies on C57BL/6J-Iba1-eGFP mice (a gene encoding enhanced green fluorescence protein under the *iba1* promoter) using in vivo 2-photon microscopy showed that microglial cells in the neocortex exhibit age-related changes: increases of a soma volume, shortening of microglial processes, reduction of motility of processes, and changes in tissue distribution [29]. Another mouse strain CD11b-CreERT2;R26-tdTomato expresses the red fluorophore tdTomato under the control of the *cd11b* gene promoter. In vivo single-cell imaging in triple-transgenic CD11b-CreERT2;R26-tdTomato;APPPS1 mice (an Alzheimer disease model, double transgenic mice expressing a chimeric mouse/human amyloid precursor protein (Mo/HuAPP695swe) and a mutant human presenilin 1 (PS1-dE9) showed that ≈20% microglia disappear in areas without amyloid deposits over the 6-month imaging period, whereas the microglia loss in the wild-type mice was ~13% over the same imaging period. The results suggested that the increase in microglia around amyloid deposits results from microglial proliferation in plaque-free areas and migration toward the plaques. The newly appearing microglia were derived from the division of resident microglial cells [30].

### 1.2. Microglia Heterogeneity and Function in Health and Disease

Microglia play a pivotal role in brain development, immune defense, and the maintenance of CNS homeostasis [31]. Microglia, with ramified and motile processes, surveil the brain parenchyma for dysfunction, infection, or damage [32]. Microglia undergo morphological changes under pathological conditions that can be categorized and quantified for parameters such as soma size, cell ramification, branching complexity, and shape. Generation of transgenic *Cx3cr1*^+*/GFP*^ mice with a one allele coding for the CX3C chemokine receptor (Cxc3cr1) replaced with GFP (green fluorescence protein) allowed to study morphological changes of microglia. For example, severe ischemia in brain slices from *Cx3cr1*^+*/GFP*^ transgenic mice leads to pronounced de-ramification and the appearance of amoeboid-shaped cells [33].

The detection of pathogen-derived signals initiates microglial responses that on one hand instigate inflammation, but on the other hand attempt to resolve the injury, protect the CNS from the consequences of inflammation, and support tissue repair and remodeling [31,34,35]. The oversimplified classification, which divides microglia into the M1 inflammatory and the M2 pro-regenerative macrophages, fails to explain the diversity of myeloid subpopulations in the diseased brain. There were attempts to attribute both functions to microglia and the prevalence of each functional subpopulation to severity of the brain damage [36]. However, in diseases that affect the integrity of the blood–brain barrier, there are considerable increases in a number of macrophages due to influx of peripheral immune cells. The pressing question in the field is whether BM-derived monocytes accumulate in the brain and how they function under pathological conditions.

Immunomagnetic sorting (MACS) or fluorescence activated cell sorting (FACS) of CD11b^+^ cells with low or high CD45 expression allowed distinguishing between CD11b^+^CD45^lo^ (microglia) and CD11b^+^CD45^hi^ (BM-macrophages) cells from rodent diseased brains [37]. Transcriptomic analyses of immunosorted microglia and macrophages from the rat ischemic brains showed the pro-inflammatory phenotype of microglia over the course of ischemia and the transient influx of the pro-regenerative macrophages into the ischemic brains [38]. With the use of chimeric mice, in which CX3CR1-GFP- monocytes were transplanted to wild-type chimeras, researchers demonstrated distinct contributions of immune cells to the brain maintenance and repair, which led to recognizing distinct roles of microglia and infiltrating BM-derived macrophages [39]. This notion has been supported by the results of conditional ablation of the BM-derived macrophages using the CD11c-DTR system, expressing the human diphtheria toxin receptor (DTR) from the *CD11c* promoter [40]. The depletion of monocytes/macrophages in CD11b-DTR transgenic mice increased the ischemic lesions and intensified the expression of the inflammatory M1 phenotype markers in CD11b+ cells [41]. The ablation of macrophages in transgenic CD11b-DTR mice had no impact on unilateral traumatic injury lesions, but it led to increases in the expression of pro-inflammatory genes in both hemispheres [42]. Other studies demonstrated that CCR2^+^Ly-6C^hi^ inflammatory monocytes are rapidly recruited to the CNS of experimental autoimmune encephalomyelitis (EAE) mice, and they are instrumental for the effector phase of disease. The selective depletion of this monocyte subpopulation with an antibody neutralizing CCR2 strongly reduced disease symptoms [43].

Microglia are present at the retina, express the CX3C chemokine receptor 1 (CX3CR1), and undergo morphological activation after corneal injury. Fate-mapping using CX3CR1^+/EGFP^::CCR2^+/RFP^ reporter mice and BM chimeras confirmed that peripheral monocytes/macrophages do not enter into retina under physiological conditions [44]. When busulfan-induced myelodepletion was followed by BM transplantation, peripheral CCR2^+^ CX3CR1^+^ monocytes migrated to the optic nerve but not to the retina under steady-state conditions. Ocular injury led to population of the retina by peripheral CCR2^+^ CX3CR1^+^ monocytes that differentiated to microglia-like CCR2^−^ CX3CR1^+^ cells. Increased monocyte/macrophage trafficking causes microglia activation and elevation of inflammation [44]. After the depletion of microglia with CSF1R inhibitor (PLX5622), even in the absence of ocular injury, peripheral monocytes repopulated the retina. After ocular injury, the engrafted peripheral monocytes were resistant to CSF1R inhibitor and retained a pro-inflammatory phenotype, expressing high levels of MHC-II, interleukin 1 beta (IL-1β), and tumor necrosis factor α(TNF-α) twenty weeks after the injury [45].

All the results point to the heterogeneity of myeloid infiltrates in CNS lesions and the distinct functions of microglia and macrophages [46]. A major challenge in the functional analysis of microglia in diseases is the lack of good experimental systems that allow discriminating between microglia and monocyte-derived macrophages. While the depletion of microglia or macrophages with CD11b-based approaches and other myeloid marker genes provided some interesting clues, these models lack microglial specificity and target other CNS and peripheral cell types. Malignant glioma, a common brain tumor, is another CNS disease, in which a complexity of a tumor microenvironment, infiltrated with numerous immune cells consisting up to 30% of a tumor mass, presents a challenge. Immune cells infiltrating gliomas consist of microglia, BM-derived monocytes, granulocytes, myeloid-derived suppressor cells (MDSCs), and T lymphocytes. The predominant population is glioma-associated microglia and macrophages (GAMs) that accumulate in high-grade gliomas, and their numbers inversely correlate with a patient survival (reviewed in [47,48]). Although GAMs have a few innate immune functions intact, their ability to secrete cytokines and upregulate co-stimulatory molecules is not sufficient to initiate anti-tumor immune responses. Moreover, tumor-reprogrammed GAMs release immunosuppressive cytokines and chemokines blocking anti-tumor responses. GAMs may contribute to tumor progression in different ways: by promoting genetic instability, supporting cancer stem cells, priming invasion, and taming anti-tumor immunity (reviewed in [47,48]). Cell transplantation studies using head-protected irradiation have shown a massive influx of the donor-derived myeloid cells in murine gliomas [49]. Peripheral monocytes/macrophages were detected in gliomas by flow cytometry as Ly6C/MHCII/MerTK/CD64 positive cells [50]. Immunofluorescence studies of platelet-derived growth factor B (PDGFB)-driven gliomas and GL261 gliomas developing in Cx3cr1GFP^+^Ccr2RFP^+^ transgenic mice showed a different location of cells: macrophages (GFP^+^RFP^+^) are predominant in the tumor core, while microglia (GFP^+^RFP^-^) accumulate in a tumor periphery [51].

We and others have addressed the issue of heterogeneity and functions of GAMs (usually isolated as CD11b^+^ cells) using bulk transcriptomic approaches, but those studies demonstrated conflicting results [52]. While our studies suggested the pro-tumor phenotype (resembling the wound healing and immunosuppressive M2 phenotype) of GAMs from rodent and human gliomas [53,54,55], other studies showed mixed M1/M2-like phenotypes of GAMs from human glioblastomas [56], a mixed inflammatory/anti-tumor phenotype in mouse GL261 gliomas [57], or M0 phenotype in human glioblastomas [58]. Lineage tracing studies provided direct evidence for the contribution of BM-derived macrophages to GAMs in murine gliomas [59]. However, the pressing issue of a cell type and functional diversity of GAMs could not be resolved with classical methods.

The heterogeneity of microglia in various regions, distinctive functions, and contribution of BM-derived monocytes/macrophages and specific roles of these subpopulations in health and disease could not be solved with the traditional methods. Single-cell studies provided a breakthrough and novel insights into the diversity of microglia in health and disease. Mass cytometry or CyTOF (Cytometry by Time-Of-Flight) is a novel platform for high-dimensional phenotypic and functional analysis of single cells. This system uses elemental metal isotopes conjugated to monoclonal antibodies to evaluate over 40 parameters simultaneously on individual cells with minimal overlap between channels [60]. Single-cell RNA sequencing (scRNA-seq) permits determining the entire transcriptome of thousands of individual cells. In recent years, scRNA-seq has been used to study several different tissues and organs, both during development and at a fixed point in time [61].

Here, we summarize the recent findings originated from studies employing single-cell technologies to study the diversity and functions of microglia in healthy and diseased brains. The studies were selected from the Pubmed database with the use of keyword combinations: “microglia” and “single-cell”; “myeloid”, “single-cell”, and “brain” within a title (single-cell, microglia) or an abstract or title (myeloid, brain). The total of 26 original articles published in years 2017–2021 were included (Table 1).

## 2. Microglia in Developing and Adult CNS—Insights from Single-Cell Omics Studies

### 2.1. Developmental Diversity of Microglia

Microglial precursors colonize the brain early in the prenatal life [15] and during maturation undergo substantial transcriptomic changes. Two independent scRNA-seq analyses of microglia (purified by FACS with the markers CD45, CD11b, and Cx3cr1) throughout mouse developmental stages demonstrated specific developmental profiles of gene expression [74,84]. While the expression of some microglial genes such as Fc receptor-like S (*Fcrls*), triggering receptor expressed on myeloid cells 2 (*Trem2*)*,* and Complement C1q A (*C1qa*) was uniform in microglia derived from all the investigated stages [74], *Tmem119* [74,84], *P2ry12*, *Cx3cr1* [74], *Selplg*, and *Slc2a5* [84] were highly expressed in microglia in adult mice, but they showed low expression in microglia from the developing brain. Embryonic and early postnatal microglia had increased expression of *Arg1*, *Rm2*, *Ube2c*, *Cenpa*, *Fabp5*, *Spp1*, *Hmox1,* and *Ms4a7*, whereas juvenile and adult microglia expressed the canonical genes but did not show any stage-specific gene expression [74]. Consistently, scRNA-seq of microglia from adult mouse brains showed lower expression of *Tmem119* and *P2ry12* in the cells identified as the microglial precursors compared to other microglial clusters [70], indicating that some of the canonical microglia genes are expressed only in the mature stage.

The specific microglia subpopulation was distinguished among microglial cells in early postnatal, pre-myelinated brains. These cells were Spp1^+^Gpnmb^+^Clec7a^+^, presented an ameboid shape, showed a high level of lysosomal membrane proteins Lamp1 and Cd68, and resided within the axon tracts of the corpus callosum and cerebellar white matter [63,74]. Those features may indicate their increased phagocytic activity and a potential involvement in the synaptic pruning. Similarly, Masuda et al. (2019) identified a subpopulation showing enhanced expression of the lysosomal cathepsin encoding genes *Ctsb*, *Ctsd*, and *Lamp1* among microglia in the embryonic brain [84]. The results are summarized in the Figure 1.

Proliferative microglia that are characterized by the enrichment of cell-cycle pathways were predominant in the prenatal and early postnatal stages but sparse in the adult brain [74]. These observations corroborate the currently accepted view that microglia are long-living cells and their multiplication rate is low in the adulthood, under steady-state conditions. Nevertheless, adult microglia are capable of reactivating the proliferation program. Upon microglia depletion with the CSF1R selective inhibitor (PLX5622), the remaining cells (~2% of the original population) upregulated the cell cycle genes (*Cdk1*, *MKi67)*, migration (*Cd36)*, and cell-survival related genes (*Bcl2a1a*, *Bcl2a1d)*, and reconstituted the whole population within 7 days [85]. Multiple lineage tracing analyses demonstrated that repopulated microglia are not derived from astrocytes, oligodendrocyte precursor cells, excitatory neurons, GABAergic neurons or catecholaminergic neurons, or nestin positive progenitors, but solely from the residual microglia. Parabiosis experiments excluded that newly generated microglia originate from BM monocytes/macrophages. Nevertheless, as CSF1R inhibition by PLX5622 causes not only microglia depletion but also reduces a number of peripheral monocytes, hematopoietic progenitor, and stem cells [86], it cannot be excluded that the depletion of BM-derived cells affects their contribution to CNS macrophage repopulation. A transcriptomic analysis of sorted CD11b^+^CD45^low^ cells showed that the repopulated microglia induced similar transcriptomic changes in response to the lipopolysaccharide (LPS) challenge; however, their gene expression profiles at day 21 were distinct from the microglia of naive mice [85]. It is unknown whether those cells present the same heterogeneity and regional specialization as the original population.

The application of scRNA-seq on myeloid cells across species—human, macaque, marmoset, sheep, mouse, hamster, chicken, zebrafish—revealed the evolutionary conserved, as well as species-specific gene expression patterns [65]. Microglial genes that were highly expressed and conserved across all species include genes involved in microglia development (*Sp1*, *Irf8*, *Tgfbr2*, *Csf1r*), genes linked with homeostasis (*C1qc*, *P2ry12*), and genes encoding lysosomal hydrolases cathepsines (*Cst3*, *Ctsa*, *Ctss*, *Ctsb*, *Ctsh*, *Ctsc*, *Ctsz*, and *Hexa*). The species-specific signature indicated that *Spp1*, *C3*, and *Vsig* are expressed at highest level in large mammals, whereas the increased expression of *Ccr5*, *Fcrls*, and *Siglech* occurs specifically across various mouse strains [65].

### 2.2. Functional Specialization of Microglia under Physiological Conditions

Mass cytometry with a panel of 43-heavy metal isotope-tagged surface antibodies complemented by 22-color fluorescence cytometry of brain immune cells showed that microglia are the predominant population in the brain of 8-week-old mice [20]. BAMs, the second abundant population of the brain immune cells (~10%) express similar surface protein markers as microglia, but they can be distinguished by the expression of surface proteins CD206, CD3820, and transcriptional markers *Apoe*, *Ms4a7*, *Mrc1*, and *Pf4* [70,76,80], although they show substantial regional differences [76,80]. Other minor immune cell populations (<3%) found in the healthy brain include neutrophils, monocytes, dendritic cells (DCs), natural killer (NK) cells, natural killer T (NKT) cells, T-cells, B-cells, and mast cells [20,62]. The scRNA-seq analysis of sorted CD11b^+^ cells from brains of adult mice confirmed that microglia are the predominant population of myeloid cells in the brain and similar percentages of BAMs, NK cells, and DCs have been detected [70]. Microglia in the adult brain represent homeostatic microglia characterized by the high expression of core microglial genes *P2ry12*, *Tmem119*, and *Olfml3* [63,70].

A recent scRNA-seq analysis of CD11b^+^ cells from adult, naïve mice demonstrated the presence of distinct microglial cell clusters, revealing an unexpected cell heterogeneity [70]. We found several subgroups among microglia (defined as clusters) that differ in gene expression profiles. The predominant subpopulation was characterized by high expression of microglial genes (*Crybb1*, *Cst3*, *P2ry12*, *Pros1*). Another group was characterized by high expression of immediate early genes (*Jun*, *Junb*, *Jund*, *Fos*, *Egr1*, *Klf6*, *Aft3*) encoding transcription factors, and it may encompass a subpopulation of transcriptionally active cells [70].

The microglial subpopulation enriched in cells expressing immediate early genes was also identified by Li et al. (2019) [63]. The authors suggested that these genes are rapidly upregulated in response to external stimuli, and their expression could be induced by the sorting procedure, which was supported by the absence of Fos and Egr1-positive microglia in the tissue sections. Thus, it remains to be solved whether the transcriptionally active microglia represent a subpopulation predisposed to rapidly react to homeostasis disturbances or constitutes an experimental artifact. The third group exhibited high expression of genes coding for a signaling inhibitor Bmp2k, transcriptional repressors: Bhlhe41, Ncoa3, and Notch2 [70]. Ncoa3 in association with nuclear receptors represses the expression of inflammation mediators and activates genes encoding anti-inflammatory mediators [87]. A subgroup of cells expressing classical microglia genes had also high expression of genes characteristic for a premature state (*Csf1*, *Mcm5*, *Ifit3*) [88]. These microglia upregulated genes encoding a cysteine protease inhibitor (*Cst7*), cytokines (*Mif* and *Csf1*), chemokines (*Ccl12*, *Ccl3*, *Ccl4*), genes involved in a response to interferon (*Ifit1*, *Ifit3*, *Ifit3b*, *Ifitm3*, *Irf7*), genes implicated in ISG-ylation (*Isg15*, *Usp18*), and may represent surveying cells fitted to rapidly respond to any dysfunction [70]. This diversity of microglia may reflect different functional states, specific subpopulations, or regional heterogeneity.

The confirmation of microglia diversity in human studies came from scRNA-seq of FACS-sorted CD45^+^ cells isolated from cortices of surgically resected human brain tissues without histological evidence of CNS pathology [84]. The authors identified several groups of human microglia: the first cluster was enriched in the canonical microglia genes *TMEM119*, *P2RY12*, *CX3CR1*, *P2RY13*, and *SLC2A5*, the second expressed *CST3*, and the third was characterized by high levels of expression of the chemokine genes *CCL2* and *CCL4*, and the zinc finger transcription factors *EGR2* and *EGR3*. The authors concluded that the homeostatic human microglia have distinct gene expression patterns from the pathological state microglia, and their expression profiles partially overlap with those of homeostatic adult mouse microglia [84].

### 2.3. Regional Specialization

The brain environment is not uniform, as different brain structures consist of various neuronal subtypes, neurotransmitter profiles, and are differentially exposed to external stimuli, which might influence the regional specialization of microglia. Single-cell resolution studies revealed different types of neurons and other neuronal cells in different nuclei or areas of CNS [89]. Numerous studies combining cell staining and sorting, advanced morphology analyses, and transcriptomic approaches demonstrated that microglia differ in their cell number, regional distribution, and molecular signatures as well as relevant functions in different mouse brain areas (reviewed in [90]). However, the spatial heterogeneity of microglia have been verified only recently. The results of the first genome-wide bulk study demonstrated distinct transcriptional profiles of cortical and striatal microglia, as compared to microglia derived from cerebellum and hippocampus, and they indicated the more immune-vigilant state of the latter [91]. The scRNA-seq analysis of microglial cells from multiple anatomical regions of the embryonic (embryonic day (E)16.5), juvenile (3 weeks), and adult (16 weeks) mouse CNS revealed distinct distribution of microglia clusters across different regions of the CNS during embryonic and postnatal stages [84]. The important differences were validated by immunocytochemistry. For example, Cst3^+^Sparc^+^Iba1^+^ microglia (Cst3, cystatin 3; Sparc, secreted protein acidic and rich in cysteine) were present in the postnatal brains but absent in the embryonic forebrains. In the juvenile cortex, most microglia expressed *Cst3* and *Sparc*, while those cells were diminished in the adult cortex. Such developmental changes were not detected in the cerebellum [84].

Phenotypes of human CNS-resident microglia (huMG) isolated from different brain regions, peripheral blood mononuclear cells (PBMCs), and immune cells from the cerebrospinal fluid (CSF) of fresh epileptic and postmortem brain tissues of nine donors were analyzed with the CyTOF panel of 57 markers [64]. The authors investigated huMG from five brain regions: subventricular zone (SVZ), thalamus, cerebellum, temporal lobe, and frontal lobe. They defined the unique huMG signature that distinguishes microglial cells from mononuclear cells: TMEM119+, P2Y12+, CD64hi, CX3CR1hi, TGF β1hi, TREM2hi, CD115hi, CCR5hi, CD32hi, CD172ahi, CD91hi, CD44-/lo, CCR2-/lo, CD45-/lo, CD206-/lo, CD163-/lo, and CD274 (PD-L1)-/lo. Among microglia, they detected three subpopulations that demonstrate regional variability and could be distinguished by different levels of the following markers: CD11c, C206, CD45, CD64, CD68, CX3CR1, HLA-DR, and IRF8. One of the identified subpopulations was composed predominantly by microglia from SVZ and thalamus, and it contained microglia overexpressing proliferation-related proteins: CYCLIN, CYCLIN B, and KI-67. The other two microglial clusters originated from frontal and temporal lobes, and both upregulated CD206, but differed by the levels of CD64 and EMR1 [64]. EMR1 was expressed at the higher level in the white matter-derived microglia that expressed higher levels of human leukocyte antigen DR (HLA-DR), Apolipoprotein E (APOE), and a lysosomal marker CD68, compared to the microglia from the gray matter [71].

In contrast, a single-cell RNA-seq study of microglia from four mouse brain regions (cortex, cerebellum, hippocampus, striatum) demonstrated a low number of differentially expressed genes and a high correlation of the expression profiles between regions [63]. Although the study was designed to detect even low expression genes, as the sequencing depth was around 20 times higher than usual (1 million reads/cell), the number of the sequenced cells was rather low (696 cells from the adult brain), raising a question of whether a sufficient representation of regional subpopulation was achieved.

The inconsistent results of high-resolution studies of microglia from human and mouse brain regions might be due to the higher degree of functional diversity of the human brain, as microglia diversity was more pronounced in human brains as compared to mouse brains of various strains [65]. It cannot be excluded that human microglia heterogeneity is associated with aging or microglial polarization within the epileptic brains, as post-mortem samples and surgery samples from the epileptic brains were used as sources of the human brain tissue.

### 2.4. Sex-Related Differences in Microglia

Microglia plays multiple roles in the brain development and may contribute to the sexual differentiation of the brain. Sex-related differences in a number of microglia and a fraction of ameboid microglial cells have been found in the preoptic area that is sexually dimorphic, and microglia were found to be essential for the process of brain masculinization mediated by prostaglandin 2 [92]. A bulk RNA seq study showed that microglial transcriptome is distinct in mice of different sex [93,94], and the sex-specific gene expression is retained upon transplantation to the opposite sex [93].

Interestingly, a scRNA-seq study that compared the transcriptomes of male and female microglia across developmental stages did not find major differences in female and male cell distribution across defined clusters, although CD74+Ccl24+Arg1+ microglia comprising 0.5% of all microglial cells were more frequent in female brains [74]. Sex-related differences were found in the microglia of adult mice. Male microglia showed an enrichment of inflammation and antigen presentation-related genes, and they were more reactive to the ATP stimulation [94]. In contrast, female microglia exhibited a higher neuroprotective capacity, as they better resolved damage after ischemia [93]. At a single-cell level, male microglia were found to express MHCII and CD74 genes at a higher level than female microglia in experimental murine gliomas and human gliomas [70].

Those few reports indicate that sex-related differences may be important for microglia functions. The prevalence of many immune system-related brain pathologies differs between women and men: women are more susceptible to autoimmune diseases, and men have a higher risk of death for a majority of malignant cancers [95] but also present better therapeutic outcomes in the immune checkpoint inhibitor therapy of various cancers [96]. More studies on microglia in both sexes are required to comprehend the importance of a sex-related variance, but this issue remains largely unexplored. Ensuring a proper representation of both sexes in further studies of microglial function in diseases may expand our understanding of sex-differences in the prevalence and outcome of brain pathologies.

## 3. Microglia in Neurological Diseases—Insights from Single-Cell Omics

### 3.1. Glioma and Brain Metastasis-Associated Microglia

Histopathological and flow cytometry studies showed an accumulation of microglia, monocytes/macrophages, and other myeloid cells in primary and metastatic brain tumors [55]. However, more detailed characteristics of those immune infiltrates were hampered by the lack of reliable markers of specific subpopulations. The first scRNA-seq analysis of myeloid cells that infiltrate human brain tumors was carried out on isocitrate dehydrogenase (IDH)-mutant adult glioblastomas. The authors found a phenotypic spectrum ranging from microglia- to macrophage-like cells based on the gradual change of expression of microglia and macrophage markers [67]]. The *IDH* mutation status emerges as an important factor determining the composition and functions of immune cells in gliomas. The immune quiescence of IDH-mut gliomas has been observed in human cohorts, which show lower accumulation of tumor-infiltrating lymphocytes (TILs) and lower programmed death-ligand 1 (PD-L1) expression [97]. In the mouse platelet-derived growth factor (PDGF)-driven IDH1-mut and IDH1-wt gliomas, numbers of tumor-associated monocytes, macrophages, and polymorphonuclear leukocytes were significantly reduced, while numbers of microglia and CD4+ or CD8+ lymphocytes were not significantly changed [98]. The reduced leukocyte infiltration could result from the lowered expression of cytokines observed in the IDH-mut mouse tumors [98].

Patients bearing IDH-mutant tumors have improved prognoses compared to those with IDH wild-type tumors, despite the fact that the oncometabolite (R)-2-hydroxyglutarate (R-2-HG) inhibits CD8+ T-cell recruitment and affects the antitumor immunity [99]. It was shown that the introduction of IDH mutant or treatment with the oncometabolite R-2-HG resulted in reduced levels of a chemokine CXCL10 and its regulator STAT1 in the syngeneic mouse glioma models. Downregulation of those factors was associated with decreased T-cell recruitment [100]. A better prognosis of patients with IDH-mut gliomas could be in fact connected to the lower rate of leukocyte infiltration, as the number of tumor-supportive GAMs is reduced.

Another scRNA-seq profiling of CD11b+ cells from high-grade and low-grade gliomas has shown a similar continuum from microglia-high to monocyte/macrophage-high marker gene expression; however, the authors did not provide annotation on a IDH status in their analysis [68]. They identified tumor-activated microglia and putative monocytes/macrophages, and they demonstrated distinct signatures using marker genes from murine glioma models [59]. The authors concluded that BM-derived GAMs upregulate M2 phenotype-associated immunosuppressive cytokines and markers of an oxidative metabolism that are characteristic of the M2 phenotype [68].

In a recent study two CyTOF panels measuring 74 parameters of immune cells were applied to analyze leukocytes from human gliomas and brain metastases (BrM) in the tumor microenvironment (TME) [69]. The authors defined subpopulations with the following parameters: CNS-resident and invading monocytes/macrophages (CD64+, CD11c^+^, and CD11b^+^), neutrophils (CD66b^+^ and CD16^+^), two subsets of dendritic cells (CD141^+^ and CADM1^+^ for cDC1 and CD1c^+^ for cDC2), T-cells (CD3^+^), NK cells (CD56^+^CD16^+^), B-cells (CD19^+^ and HLA-DR^+^), and plasma cells (CD19^+^ and CD38^high^. They showed that monocytes/macrophages constitute one-quarter of the myeloid immune infiltrate in IDH1-wt gliomas and one-half in brain metastases, but they are rare in human IDH-mut gliomas [69]. Microglia in IDH1-mut gliomas expressed similar levels of HLA-DR as microglia in IDH1-wt gliomas and BrM; however, only in IDH1-wt gliomas did these cells upregulate CD14 and CD64, which is a sign of their activation. The re-analysis of a population of the ITGA4/CD49d expressing cells (the integrin alpha 4 was previously reported to specifically mark CNS-invading macrophages [59]) demonstrated that glioma infiltrating monocytes and macrophages show a gene expression trajectory consistent with a monocyte-to-macrophage transition in the TME [69]. The frequency of monocyte-to-macrophage ratio varied depending on the tumor type. CNS-invading leukocytes in the IDH1-mut gliomas were predominantly composed of monocytes and lower frequencies of macrophages. BrM exhibited a higher infiltration of leukocytes than gliomas and showed the presence of the monocyte-derived macrophages (MDM) subpopulation expressing CD206, CD209, CD169, CD163, and high levels of CD38, PD-L1, and PD-L2 [69].

Recent CyTOF and scRNA-seq studies of glioma TME were based on a higher number of cells, and they showed a more reliable separation of microglia and peripheral myeloid cells [69,70]. CD49d was first proposed by Bowman and coworkers as a marker of MDMs in experimental murine gliomas [59], and it was confirmed as a good surface marker of monocytes and MDMs in human brain tumors [69,101]. Tmem119 was proposed as a microglia marker by Bennett and coworkers [102], and its utility was confirmed in murine experimental gliomas [70]. Interestingly, the single-cell studies indicate that the level of CD45 protein allows a good separation of microglia and MDMs [69,70], although the use of CD45 as a marker was criticized due to its expression increase in microglia in TME [49].

The stratification of glioblastoma multiforme (GBM) patients according to an MDM marker CD163 showed that the high CD163 level correlates with worse overall patient survival, whereas the level of microglial marker CX3CR1 showed no correlation [69]. This is in accordance with early scRNA-seq studies on human gliomas indicating that the fraction of MDMs increases with a tumor grade [67], whereas frequencies of microglia do not differ between low-grade and high-grade gliomas as shown with the cell type identity score based on RNA-seq data from the cancer genome atlas (TCGA) [68].

Different single-cell studies showed consistently that MDMs tend to localize within the tumor core, whereas microglia reside mainly on the tumor edge and in the adjacent brain parenchyma [59,66,69,70]. In human glioma samples, myeloid cells isolated from the tumor periphery showed an increased expression of genes coding for cytokines (*CCL3*, *CCL4*, T*NF*) and the pro-inflammatory interleukins (*IL1B*, *IL1A*, *IL6-R*), whereas immune cells isolated from the tumor core upregulated genes involved in angiogenesis (*VEGFA*, *VEGFB*) and encoding inhibitors of pro-inflammatory cytokines (*IL1RN*) [66], indicating the tumor-supportive phenotype of GAMs within a tumor core.

A main difficulty in discriminating tumor-activated microglia and MDMs in experimental gliomas stems from the fact that both subpopulations upregulate similar transcriptional networks, although the degree of activation may vary in a specific subset. In murine gliomas at the asymptotic phase, MDMs upregulated *Il-1b* that encodes an inflammatory cytokine, along with genes encoding inflammatory cytokine inhibitors *Il1rn* and *Il18*, and there was a subset with a high expression of the *Cd274* mRNA encoding an immune-checkpoint inhibitor (*PD-L1*). In contrast, tumor-activated microglia specifically upregulated *Ccl12*—a cytokine attracting monocytes and lymphocytes [70].

Summarizing, single-cell resolution studies of rodent and human gliomas have shown a predominant monocyte/macrophage accumulation and localization in the tumor core, monocyte-to-macrophage transition, a strong impact of MDM accumulation on patient’s survival, and the expression of immune checkpoint inhibitors by MDMs. These characteristics point to their immunomodulatory and immunosuppressive roles in brain tumor progression. Microglia rather decrease their signature genes in human gliomas, and being at the tumor invasive edge facilitates diffusive tumor growth in the brain parenchyma. Patient stratification based on the composition of the immune infiltrates may be informative in the selection of the best immunotherapy approach. Further research is needed to determine whether a personalized therapy tailored to a specific composition of the immune TME may increase patient survival.

### 3.2. Microglia Functional Diversity during Aging and Neurodegeneration

Aging may profoundly affect the brain function and constitutes a major risk factor for most neurodegenerative disorders, including dementia and Alzheimer’s disease (AD). Mass cytometry with a 43-heavy metal isotope-tagged surface antibody panel combined with location information from immunohistochemistry (IHC), 22-color fluorescence cytometry, and reporter and fate-mapping systems allowed analyzing changes of myeloid cells in 2-month-old versus 1.5-year-old mice [20]. Aged mice showed alterations of the microglia phenotype, as a subset of the microglia with high levels of phagocytosis markers CD11c and CD14, antigen presentation protein complex MHCII, and increased CD44, CD86, MHCII, and PD-L1 appeared in aged mice compared to adult animals [20]. A similar phenotypic signature was detected in a subset of CD11c^hi^ microglia in AD-prone mice, which was found in the surroundings of amyloid precursor protein (APP) plaques in both amyloid precursor protein /presenilin 1 (APP/PS1) and 5xFAD [72] mouse models of AD. Additionally, in aged mice, differences in the frequencies of other immune cell types have been noted: a number of T-cells, MHCII^+^ BAMs increased, whereas MHCII^-^ BAMs, NKT, and pDCs were less frequent [20,73].

Keren-Shaul and coworkers [72]] described a novel type of microglia associated with neurodegenerative diseases—disease-associated microglia (DAM). ScRNA-seq analysis of DAM in Tg-AD and triggering receptor expressed on myeloid cells 2 (Trem2)-/- Tg-AD mice demonstrated that differentiation to DAM is a two-step process, in which Trem2-independent activation involves downregulation of the core microglia genes *P2ry12*, *Cx3cr1* and *Tmem119*, and upregulation of *Tyrbp*, *Apoe*, *B2m*, *Ctsb*, and *Ctsd*, whereas a late-stage DAM upregulated *Spp1*, *Axl*, *Ccl6*, *H2-D1,* and genes involved in phagocytosis and lipid metabolism: *Lpl*, *Cst7*, and *Cd9* [72]. Interestingly, microglia in the Trem2-/- 5xFAD mice expressed the early-stage DAM phenotype but failed to initiate the late-stage DAM activation. Mutations in the *TREM2* gene are associated with an increased risk of AD development, and its absence accelerates the accumulation of AB plaques and neuronal loss, implying that the Trem2-dependent DAM phenotype is likely to mitigate the disease progression. A computational analysis established that microglia in the AD model display a transition from the homeostatic microglia to the DAM population as disease progresses. Immunostaining for Iba1 and Lpl (a DAM-specific gene coding for lipoprotein lipase was previously identified as a risk factor in AD) combined with histological staining using Thioflavin-S demonstrated the localization of positive cells in regions with a high density of Aβ plaques. A strong overlap between Lpl^+^ microglia and Aβ plaques was detected in AD postmortem brain samples [72].

A similar DAM profile was detected in the CK-p25/Cdk5 inducible mouse model of neurodegeneration in which the early response microglia induced *Top2a*, and the late response microglia displayed the upregulation of *Apoe*, *Axl*, *Lgals3bp,* and antigen presentation-related genes *H2-Ab1*, *H2-D1,* and *CD74* [77]. Similarly, in the model of amyotrophic lateral sclerosis (ALS) - mSOD1, microglia showed a reduction of the core microglial genes *Tmem119* and *P2ry12* [72,76] and upregulated *Trem2*, *Tyrobp*, *Lpl,* and *Cst* [72]. This is in line with the findings from the mouse models of neurodegeneration or demyelination, in which both a facial nerve axotomy and cuprizone-induced demyelination upregulated the expression of *Itgax* (CD11c), *Apoe*, *Axl*, *Igf1*, *Lyz2,* and *Gpnmb* [83,84].

RNA-seq performed on single nuclei from post-mortem frozen tissue from patients with low and high amyloid burden provided additional evidence of the AD-associated microglia phenotype [78]. A single-nucleus RNA-seq included approximately 2000 microglial cells that increased the expression of the DAM genes: *CD81*, *APOE*, *SPP1*, *CD74*, and *HLA-DRB* when compared to other microglial cells [78]. The opposing results were acquired in a recent scRNA-seq study of 11 post-mortem samples from patients with a mild cognitive impartment (MCI) and AD, and three control samples derived from the epileptic brains [79]. In line with the previous animal studies [72,77], the authors found that the expression of early DAM response genes increased in a proliferative cell cluster that appears predominant in the MCI/AD samples. However, other DAM genes (*APOE*, *LPL*, *SPP1, CD74*) were distributed across all clusters, with the highest expression in clusters enriched in cells from epilepsy-derived controls [79]. Such low consistency of the findings is surprising, but it may stem from using epilepsy-derived tissues as controls, as other studies have shown an induction of the DAM profile in epileptic subjects and a mouse model of epilepsy (see Section 3.3).

The presence of the DAM microglia phenotype is not associated with a specific cause of disease pathology (as this population has been also detected in ALS), but rather with a general program that is involved in the clearance of misfolded or aggregated proteins that accumulate in neurodegenerative diseases and during aging-induced damage. Identification of the DAM signature in aged mouse brains [72] together with the evidence from human studies reporting correlation of the expression of their human orthologs with age [77], and high expression of *SPP1*, *CTSD*, *APOE*, *LPL*, and *B2M* in microglia derived from the oldest donors (50–80 years) [71], support the notion that the DAM microglia are beneficial for neurodegenerative disorders and aging.

Notably, animal studies demonstrated that the AD-associated microglia signature is largely shared with the signature of BAMs, which was previously described by Hove and coworkers [80]. The choroid plexus BAMs showed increased expression of the phagocytosis and lipid metabolism-related genes, *Apoe*, *Cst7*, *Clec7a*, and *Lpl*, although the expression of *Apoe* is increased in all types of BAMs (reviewed in [103]). Thus, a substantial overlap between AD/aging-associated microglia and BAMs should be further explored, as a better separation of two cell populations may yield consistent results.

### 3.3. Microglia and Neuroinflammation

Neuroinflammation encompasses a broad range of brain pathologies including infection, injury, amyotrophic lateral sclerosis (ALS) and multiple sclerosis (MS). In a model of the acute inflammation induced by a systemic injection of lipopolysaccharide (LPS), scRNA-seq of CD11b^+^CD45^int^ microglia showed decreased expression of the homeostatic genes (*Tmem119*, *Siglech*, *P2ry12*, *P2ry13*, *Mef2c*), and induction of the pro-inflammatory profile, which is characterized by an increased expression of *Irf1*, *Tnf*, *Ccl2,* and *Nfkbia* [82]. Interestingly, the main population of LPS-activated microglia upregulated also genes, which were found in microglia from AD and neurodegeneration models (*Lyz2*, *Apoe*, *H2-D1*, *Fth1*) and from gliomas (*Cd52*, *Ccl12*) [82]. The results are summarized in the Figure 2.

Mass cytometry with a panel of 255 metal isotope conjugated antibodies was applied to study innate immune populations of CNS and blood samples in three mouse neuroinflammation models: experimental autoimmune encephalomyelitis (EAE, an animal model of MS), R6/2 mouse with N-terminally truncated mutant huntingtin with CAG repeat expansion (a model of Huntington disease), and mutated superoxide dismutase 1 (mSOD1, a model of ALS) [104]. The CNS-specific cell fraction encompassing microglia from healthy brains was demarcated by the CD45^low^CD11b^+^Ly6G^-^Ly6C^-^ signature, among which two microglial subtypes were distinguished: the CD39^low^CD86^-^ microglia with low levels of signaling molecules: phospho-CRE binding protein (pCREB), phospho- MAPK Activated Protein Kinase 2 (pMAPKAP), nuclear factor kappa-light-chain-enhancer of activated B cells (NF-κB), CCAAT-enhancer-binding proteins (C/EBPβ), as compared to the CD39^hi^CD86^+^ microglia population. In EAE, from the pre-symptomatic to the peak stage of the disease, the CD39^hi^CD86^+^ microglia showed induction of the signaling molecules, and a third MHCII^hi^ microglia population appeared, which was accompanied by the influx of the Ly6C^+^ monocytes. This effect was transient, as MHCII^hi^ microglia and Ly6C^+^ monocytes decreased throughout the chronic to recovery phase [104].

Multiple sclerosis is an autoimmune disease accompanied by the increased influx of monocytes and macrophages at the lesion sites. Jordao and coworkers investigated whether the accumulation of myeloid cells at the lesion sites results from the expansion of resident microglia or influx of MDMs [76]. scRNA-seq was performed on 3461 CD45+ cells isolated from different CNS compartments (including leptomeninges, perivascular space, parenchyma, and choroid plexus) in an EAE mouse model [76]. The authors found an increased number of both cell types in the brain parenchyma at the disease peak, whereas in leptomeninges and perivascular space, only the number of MDMs was increased. Both microglia and MDMs were Ki67^+^, which is suggestive of a proliferative state, especially at the disease peak, which was followed by apoptosis and reduction of the cell number in the chronic phase [76]. Several MDM populations were observed during disease progression, including monocyte-derived cells expressing MER proto-oncogene tyrosine kinase (Mertk) and Mannose receptor C-type 1 (Mrc1) or expressing zinc finger transcription factor Zbtb46 and Cd209a. The density of DCs highly increased during disease progression. DCs and monocyte-derived cells, but not microglia, were the major cells engaged in antigen presentation [76]. Time-lapse imaging of Cx3cr1CreERT2:R26tdTomato:Cd2GFP and Ccr2RFP: Cd2GFP transgenic mice showed interactions of T-cells with circulating MDMs rather than tissue-resident macrophages during neuroinflammation [76]. scRNA-seq on Tmem119^+^ cells in a model of lysolecithin (LPC)-induced demyelination revealed that the transcriptional profile of microglia from the brain lesions resembled to some extent the disease- and tumor-associated microglia [74]. A vast majority of microglia isolated from the lesions had reduced expression of the core microglial genes (*P2ry12*, *Selplg*, *Cx3cr1*) and upregulated expression of *Apoe*, *Ifitm3*, *Lgals3bp*, *Cd52*, *Bst2, Ccl12*, and *MHCI* genes [74].

Corroborating the findings from animal studies, scRNA-seq on microglial cells derived from five individuals in the early phase of MS identified three disease-associated microglial clusters, all of which exhibited the reduced expression of the core microglial genes (*TMEM119*, *P2RY12*, *P2RY13*, *CX3CR1*, *SLC2A5*) [84]. The identified three clusters were characterized by elevated expression of (1) *CTSD*, *APOC1*, *GPNMB*, *ANXA*, *LGALS1*, (2) MHCII-encoding genes and *LYZ*, and (3) *SPP1* and *LPL* [84]. Among myeloid cells isolated from CSF of relapsing-remitting MS patients, microglia showed the highest expression of *APOE*, *APOC1*, and *TREM2* as compared to monocytes and dendritic cells [75].

Microglia isolated from the human temporal cortex resected from the vicinity of the epileptogenic focus showed increased levels of *P2Y12*, *IRF8*, and *CD68* [64], and enrichment within the cell clusters that upregulated genes encoding cytokines (*CCL2*, *CCL3*, *CCL4*), anti-inflammatory interleukins (*IL-10*, *IL-4*, *IL-13*), and genes involved in antigen presentation (*CD74*, *HLA*-*DMB*,), although a majority of microglial cells from the epileptic brain belonged to the homeostatic microglia population [79]. Thus, it might be that microglia isolated from histologically non-pathological brain tissue of epileptic brain exhibits signs of polarization.

### 3.4. Microglia and Major Depressive Disorder

Major depressive disorder (MDD) is one of the common mental disorders. Recent cell type-specific methylome studies demonstrated the involvement of the innate immune system in MDD [105]. The CyTOF study utilizing a 59-marker panel on microglia from four brain regions of 11 donors (17 control samples, 19 MDD samples for all regions) provided a first insight into a microglia phenotype in MDD at the single-cell level [81]. Microglia were collected as MACS-isolated CD11b^+^ cells from post-mortem brain tissue of MDD and control cases from the subventricular zone, thalamus, temporal lobe, and frontal lobe. The antibody panel allowed distinguishing eight microglial clusters that exhibited differential distribution across brain regions. A unique microglial phenotype was observed for SVZ, in which microglia showed increased levels of HLA-DR, CD11c, and CX3CR1, as compared to other regions. The most abundant microglia cluster demonstrated significant differences between MDD and controls, as the levels of the core microglia markers purinergic receptor P2Y12 (P2RY12) and transmembrane protein 119 (TMEM119) were increased, whereas the levels of the HLA-DR and the activation marker CD68 were decreased in the MDD-derived microglia. Interestingly, levels of investigated interleukins, chemokines, or other factors frequently upregulated in DAMs e.g., TREM2, APO, AXL, and PD-L1, were expressed at the same level in MDD and control microglia. Thus, microglia in MDD seem to enhance their homeostatic functions, which is in contrast with other disease-associated microglia that show downregulation of the core microglia genes. Microglia in MDD upregulate a number of function specific factors, e.g., antigen presentation proteins, monocyte attracting cytokines, or proteins involved in phagocytosis. Human microglia expressed higher levels of the chemokine CLL5 but comparable levels of IL-1β, IL-6, TNF, CCL4, IL-10, and CCL2between MDD and control cases. In contrast to previous reports indicating increases of pro-inflammatory cytokines in MDD patients and postulating microglia-driven neuroinflammation, the authors demonstrated a non-inflammatory phenotype of microglia in MDD [81].

## 4. Microglia-Targeting Therapeutic Interventions in CNS Disorders

There were many attempts to modulate functions of microglia and switch cells from a detrimental to a pro-regenerative phenotype. Numerous compounds and treatments such as cytokines, lipid messengers, or microRNAs, as well on nutritional approaches or therapies with stem cells have been tested. Altered glucose metabolism in the cytoplasm outside mitochondria accompanied inflammatory responses of microglia. Dichloroacetate (DCA) promotes glucose metabolism and oxidative phosphorylation (OXPHOS) in mitochondria, while dimethylfumarate (DMF) promotes an antioxidant response and mitochondrial biogenesis through activation of the nuclear factor erythroid 2–related factor 2 (Nrf2) pathway. Two drugs shifting glucose metabolism toward the tricarboxylic acid (TCA) cycle/oxidative phosphorylation DCA and DMF have been approved for the treatment of MS. Metformin promoting oxidative phosphorylation through AMPK activation and peroxisome proliferator-activated receptor (PPAR)-γ agonists (thiazolidinediones) showed promising activities in taming excessive microglial activation in vitro and in EAE mice (reviewed in [106]). In vitro studies demonstrated that the metabolism of microglia is shifted from OXPHOS to aerobic glycolysis (the Warburg effect), and mitochondrial fission occurs during Aβ-induced inflammatory activation. Microglia in 5XFAD mice are metabolically defective, and the treatment with interferon-γ reversed the defective glycolytic metabolism and inflammatory responses of microglia, mitigating the pathological events in 5XFAD mice [107]. Ex vivo microglia isolated from the 5xFAD mice co-expressed the potassium channels Kv1.3 and Kir2, and the application of PAP-1 (a brain penetrant Kv1.3 blocker) mitigated the pro-inflammatory and neurotoxic microglia responses induced by amyloid-β oligomer in vitro and in hippocampal slices. Treatment of APP/PS1 transgenic mice with PAP-1 for 5 months, starting at 9 months of age, reduced numbers of CD68+ cells, decreased cerebral amyloid load, restored hippocampal LTP, and improved cognitive deficits [108]. PAP-1 and the peptide Kv1.3 blocker ShK-186 did not show systemic toxicity and are promising drugs for clinical development.

Microglial TREM2 is another potential therapeutic target. The APOE pathway mediates a switch from homeostatic microglia to neurodegenerative microglia, followed by the phagocytosis of apoptotic neurons. TREM2 controls APOE signaling, and targeting the TREM2–APOE pathway restored the homeostatic signature of microglia in APP–PS1 and SOD1 mice, and it also prevented neuronal loss in an acute model of neurodegeneration [109]. Mutations in the *TREM2* gene are associated with AD risk factors. The absence of TREM in microglia exacerbates the late, but not the early, disease progression, AB plaques accumulation, and neuronal loss [110]. It has been hypothesized that the late-stage DAMs mitigate disease progression via an increased phagocytic activity, although their induction may appear too late in the course of the disease [72]. Since the induction of the late-phase DAMs was found to be TREM2 dependent, stimulation of the microglial TREM2 in the early phase of AD could accelerate microglia activation. In fact, a recent study demonstrated that the chronic activation of TREM2 in the 5xFAD mice enhanced microglia recruitment to plaques and decreased plaque deposition as well as improved cognitive outcomes [111]. In 2020, a TREM2-activating antibody (AL002) completed the phase 1 (NCT03635047) and entered the phase II clinical trials (NCT04592874) for the treatment of AD.

Understanding the specific functions of monocytes and macrophages in malignant gliomas prompted the development of targeted approaches to inhibit or reprogram those cells. In recent years, there has been a considerable progress in immune therapies in cancer. Therapeutic agents targeting immune checkpoint interactions via PD-1/PD-L1 or CTLA4 that aim to reactivate T-cell mediated immunity have been successful in a variety of cancers e.g., melanoma and non-small cell lung cancer. Unfortunately, anti-PD-1 mono- and combination therapies failed to increase glioma patient survival [112], which is likely due to the immunosuppressive TME preventing the effective infiltration of cytotoxic T-cells. Tumor-associated monocytes and macrophages were found to be a predominant cell population within the brain tumor microenvironment, although the composition of the immune landscape varies depending on the tumor type [69]. The number of monocytes and macrophages within a tumor negatively correlates with patient’s survival, while there is no such correlation for microglia [68]. Monocyte and MDMs display a stronger activation than microglia [70], and these features make MDMs a potential therapeutic target in gliomas and other brain tumors (including brain metastases) [113]. Pharmacological depletion/inhibition of GAMs, encompassing both microglia and MDMs, in murine gliomas impaired tumor growth [114,115]. Microglia and MDMs depend on CSF1R signaling for survival. A CSF-1R inhibitor (BLZ945) increased survival in a mouse proneural GBM model and regressed established tumors [116]. However, BLZ945 treatment decreased percentages of CD11b^+^Ly6G^−^ cells in blood, while numbers of GAMs in tumors were not reduced, which was attributed to glioma-secreted factors including granulocyte-macrophage colony stimulating factor (GM-CSF) and interferon-γ supporting GAM survival. Treatment with a CSF-1R inhibitor slowed the intracranial growth of patient-derived glioma xenografts [116]. Another CSF-1R inhibitor, PLX3397 (Pexidartinib), blocked growth of the murine proneural gliomas. The inhibitor blocked the polarization of GAMs in vitro and restored the sensitivity of glioma cells to tyrosine kinase inhibitors in preclinical studies in vivo [117]. After promising pre-clinical trials, PLX3397 was evaluated in a phase II clinical trial on patients with recurrent GBM. The drug was well tolerated and showed good CNS penetration. Despite reducing numbers of CD14^lo^/CD16^+^ monocytes in patient’s plasma, the treatment showed no anti-glioma efficacy [118]. Poor responses of glioma patients could be due to the high expression of GM-CSF by glioma cells that stimulates macrophage proliferation and thus prevents the depletion of glioma-associated macrophages [116].

A promising target is CD49d, which is a surface protein that is uniformly expressed by monocytes and MDMs in the glioma microenvironment [59,90,91]. CD49d is an α4 integrin that is also expressed on a surface of lymphocytes and endothelial vessels, and it plays a critical role in the recruitment of lymphocytes to the inflamed brain [119]. Anti-CD49d antibody (natalizumab) attenuated the CNS inflammation [120] and was found to be an efficient treatment of relapsing remitting MS, as it impeded disease progression and reduced relapses [119]. Natalizumab has been approved for the treatment of relapsing-remitting MS, although this therapy may lead to an increased risk of brain infection, which occurs in one out of 500 patients [119].

## 5. Conclusions

High-resolution, single-cell studies provided novel insights into the regional and functional diversity of microglia, greatly extended our understanding of microglial functions in CNS, and allowed the identification of more reliable markers to isolate or trace these cells in CNS. Particularly important findings that came for single-cell omics studies encompass identification of the AD-associated microglia phenotype. This phenotype might be masked in previous studies due to a considerable similarity between BAM and AD-microglia expression profiles. Human brain tissues used as controls for brain tumor and AD studies are rarely from healthy individuals, and they frequently originated from surgical resections of epilepsy (as an alternative of a post-mortem control tissue). The DAM phenotype is likely activated in epileptic tissues. Studying human microglia under homeostatic conditions is limited due to likely changes in microglia derived from tissues surrounding an epileptic focus.

In many CNS pathological conditions and animal disease models, a common phenomenon is the downregulation of core microglial genes: *Tmem119, P2ry12,* and *Cx3cr1*. While their expression is lower compared to homeostatic microglia, it is still at a sufficient level to discriminate microglia from bone marrow-derived monocytes and macrophages, which was shown in the brain tumor and EAE murine models [69,70,76]. The microglial phenotype induced by neuroinflammation and neurodegeneration resembles some characteristics of the premature microglia, which are found during neurodevelopment. This could reflect a fraction of the proliferating, non-fully differentiated microglia. These microglia have increased a phagocytic activity and motility.

All disease-associated microglia show an increased antigen presentation activity (defined as the increased MHCII expression) and elevated expression of monocytes and leukocytes attracting chemokines, especially Ccl12, which is uniformly expressed by tumor-activated microglia but is also found in the microglia isolated from MS patient autopsies (Figure 2).

Specific features of microglia and MDMs could be explored as therapeutic targets. Immunosuppressive MDMs constitute a promising target for glioma treatment, whereas the enhancement of a microglia phagocytotic activity via Trem2 proved its efficacy in preclinical studies and is currently in a clinical phase 2 trial. As metabolic switch seems to be a common feature of neuroinflammatory and neurotoxic microglia in many diseases, drugs shifting glucose metabolism toward the TCA cycle/oxidative phosphorylation and promoting oxidative phosphorylation through AMPK activation are of particular interest. As many of those drugs have been in clinical trials for other diseases for years and have been proved to be relatively safe, their repurposing and application in neuropathological conditions associated with increased microglia activation should provide promising results.

## Figures and Tables

**Figure 1 ijms-22-03027-f001:**
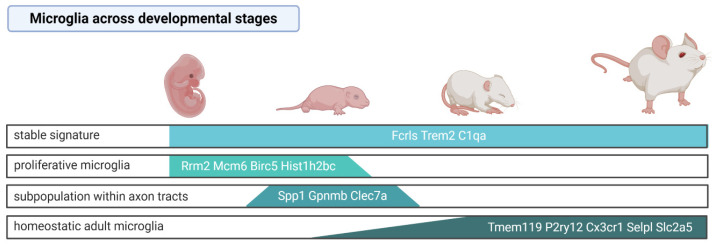
Microglia signature across mouse developmental stages based on the single-cell studies. Created with BioRender.com (accessed on 8 March 2021).

**Figure 2 ijms-22-03027-f002:**
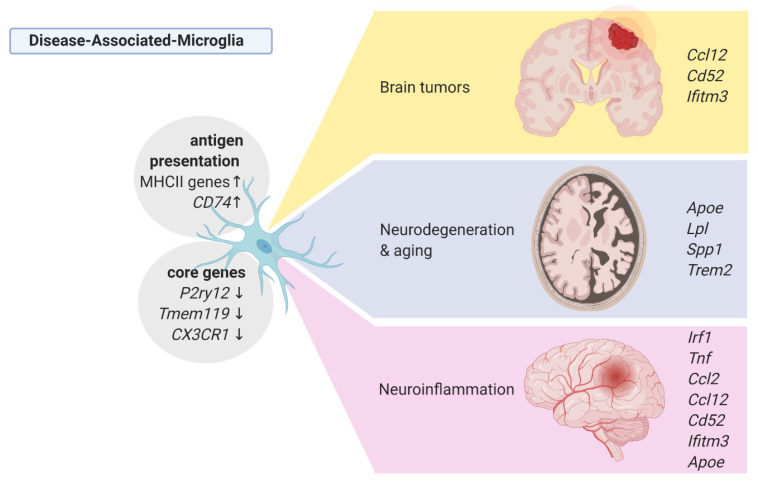
Phenotypes of the disease-associated microglia from single-cell studies. Created with BioRender.com (accessed on 8 March 2021).

**Table 1 ijms-22-03027-t001:** Summary of single-cell omics studies on the immune system cells in the brain including microglia. Blue squares indicate that study included cells from a given condition. MS—multiple sclerosis, ALS—amyotrophic lateral sclerosis, EAE—experimental autoimmune encephalitis, HD—Huntington’s disease, AD—Alzheimer’s disease, MDD—major depressive disorder.

	Healthy Brain	Epilepsy (Non-Pathological Tissue)	Brain Tumor	Aged Brain	MS	ALS	EAE	HD	AD	MDD	LPS Injection	Neurodegeneration	Microglia Depletion	Species	Number of Cells scRNA-seq	Number of CyTOF Antibodies
Korin et al. 2017 [62]														mouse	—	44
Li et al. 2019 [63]														mouse	1816	—
Böttcher et al. 2019 [64]														human	—	57
Geirsdottir et al. 2019 [65]														various	4458	—
Darmanis et al. 2017 [66]														human	3589	—
Venteicher et al. 2017 [67]														human	14,226	—
Müller et al. 2017 [68]														human	1373	—
Friebel et al. 2020 [69]														human	—	74
Ochocka et al. 2021 [70]														mouse	40,401	—
Sankowski et al. 2019 [71]														human	6411	55
Mrdjen et al.. 2018 [20]														mouse	—	43
Keren-Shaul et al. 2017 [72]														mouse	11,841	—
Dulken et al. 2019 [73]														mouse	14,685	—
Hammond et al. 2018 [74]														mouse	76,149	—
Esaulova et al. 2020 [75]														human	≈30,000	—
Ajami et al. 2020 [75]														mouse	—	255
Jordão et al. 2019 [76]														mouse	3461	—
Mathys et al. 2017 [77]														mouse	1685	—
Mathys et al. 2019 [78]														human	75,060	—
Olah et al. 2020 [79]														human	16,242	—
van Hove et al. 2019 [80]														mouse	25,384	—
Böttcher et al. 2020 [81]														human	—	59
Sousa et al. 2018 [82]														mouse	1247	—
Tay et al. 2018 [83]														mouse	944	
Masuda et al. 2019 [84]														mouse/human	5428	—
Huang et al. 2019 [85]														mouse	1194	—

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
