# Peer review of "Microglia Diversity in Healthy and Diseased Brain: Insights from Single-Cell Omics"

_ijms, 2021, doi:10.3390/ijms22063027_

Round 1
Reviewer 1 Report
This is an interesting and informative review of microglia, and outlines recent outcomes using single cell analysis.
There are few points that need to be made in this review to balance arguments. For example, there are many papers that have shown that microglia depletion is not protective but rather exacerbates neurodegeneration. Also, the use of CSF1R inhibition for microglia depletion was recently shown to have effects beyond this cell population, which may confound studied outcomes. Also, there was a recent paper in patients showing that CSF1R attenuation drives cerebrovascular pathology. All these need to be included into the review in order to balance it. Lastly, there were studies showing that various CNS and retinal pathologies can lead to monocytes infiltration into the retina. I a relevant retina study, the authors showed that these monocytes differentiate to microglia like cells, migrate to the normal microglia strata and engraft permanently into the tissue. As a result, any microglia analysis following CNS pathology my contain these “disguised” monocytes and shift the transcription profile of what is assumed to be a microglia population.
More specific comments below:
- Line 45: Explain how microglia renew themselves. Is it only through proliferation or there is a niche. What is the contribution of bone marrow cells to this?
- Line 61: It may be worth mentioning a study that used busulfan BMT to trace monocytes and differentiate microglia. In the absence of pathology, monocytes from the blood did not participate in microglia population after depletion. However, this was not the case in pathology PMID: 30442669
- Line 116: None of the fate mapping techniques are perfect and the possibility that peripheral monocytes may still infiltrate in amyloid deposits.
- Line 134: In is worth looking at the following two papers. PMID: 30442669; 30541880
- Line 168. I have the feeling that GAM was not spelled-out before use
- Lines 243-254. At least for the retina, this has been challenged. Recent reports suggest that PLX5622 affects also peripheral cells PMID: 32900927. This could have been the reason of delayed monocyte contribution in the repopulation after microglia depletion.
- Lines 568-574. The possibility that all these genes are transiently expressed by the same cells should be discussed. The gene expression signature may not be the most accurate way to associate cell lineage.
- Lines 712-727. CSF1R inhibition (genetic and PLX5622) affects non microglia cells PMID: 33203068, 32900927, 33446487. This may confound experimental data.
Some more papers worth reviewing:
- V. Bellver-Landete et al., Microglia are an essential component of the neuroprotective scar that forms after spinal cord injury. Nat. Commun. 10, 518 (2019).
- T. A. Evans et al., High-resolution intravital imaging reveals that blood-derived macrophages but not resident microglia facilitate secondary axonal dieback in traumatic spinal cord injury. Exp. Neurol. 254, 109–120 (2014).
- T. A. Wynn, K. M. Vannella, Macrophages in tissue repair, regeneration, and fibrosis.
Immunity 44, 450–462 (2016). - S. J. Karlen et al., Monocyte infiltration rather than microglia proliferation dominates the early immune response to rapid photoreceptor degeneration. J. Neuroinflammation 15, 344 (2018).
- E. I. Paschalis et al., Permanent neuroglial remodeling of the retina following infiltration of CSF1R inhibition-resistant peripheral monocytes. Proc. Natl. Acad. Sci. U.S.A. 115, E11359–E11368 (2018).
- E. I. Paschalis et al., The role of microglia and peripheral monocytes in retinal damage after corneal chemical injury. Am. J. Pathol. 188, 1580–1596 (2018).
- E. I. Paschalis et al., Microglia regulate neuroglia remodeling in various ocular and
retinal injuries. J. Immunol. 202, 539–549 (2019). - 11. N. N. Dagher et al., Colony-stimulating factor 1 receptor inhibition prevents microglial plaque association and improves cognition in 3xTg-AD mice. J. Neuroinflammation 12, 139 (2015).
- 12. D. Kokona, A. Ebneter, P. Escher, M. S. Zinkernagel, Colony-stimulating factor 1 re- ceptor inhibition prevents disruption of the blood-retina barrier during chronic in- flammation. J. Neuroinflammation 15, 340 (2018).
- 13. S. K. Halder, R. Milner, A critical role for microglia in maintaining vascular integrity in the hypoxic spinal cord. Proc. Natl. Acad. Sci. U.S.A. 116, 26029–26037 (2019).
- 14. J. M. Hutchinson, L. G. Isaacson, Elimination of microglia in mouse spinal cord alters the retrograde CNS plasticity observed following peripheral axon injury. Brain Res. 1721, 146328 (2019).
Author Response
We greatly appreciate reviewer comments and careful review of this manuscript. Their feedback was very helpful. We addressed all comments and made requested amendments to the manuscript .
Reviewer 1
There are few points that need to be made in this review to balance arguments. For example, there are many papers that have shown that microglia depletion is not protective but rather exacerbates neurodegeneration. Also, the use of CSF1R inhibition for microglia depletion was recently shown to have effects beyond this cell population, which may confound studied outcomes. Also, there was a recent paper in patients showing that CSF1R attenuation drives cerebrovascular pathology. All these need to be included into the review in order to balance it. Lastly, there were studies showing that various CNS and retinal pathologies can lead to monocytes infiltration into the retina. I a relevant retina study, the authors showed that these monocytes differentiate to microglia like cells, migrate to the normal microglia strata and engraft permanently into the tissue. As a result, any microglia analysis following CNS pathology my contain these “disguised” monocytes and shift the transcription profile of what is assumed to be a microglia population.
Ad. The reviewer rightly raised the point that the current evidence on roles of microglia in neurodegeneration is inconclusive and indeed, in some cases microglia depletion is not protective but rather exacerbates neurodegeneration. However, this is a complex issue which is confused by the fact that in some experiments depletion targeted different myeloid cells including immuno-regulatory BM-derived macrophages, therefore the precise dissection of the role of microglia is difficult or impossible. The same is true for the use of CSF1R inhibition for microglia depletion which may have effects beyond this cell population and confound outcomes. In this case it can’t be excluded that CSF1R inhibition and its impact on cerebrovascular pathology may be due to influence of BAMs rather than microglia. Single cell omics has not be been applied to solve this issue. I think discussing of such controversial issue is beyond a scope of this manuscript which is focused on single-cell studies and their contribution to understanding of microglia functions in diseases.
We thank the reviewer for turning our attention to monocyte infiltration into the retina and retinal pathologies. In fact, we focused on studies relevant to the brain and overlooked this issue. We introduced an appropriate information to the revised version.
More specific comments below:
- Line 45: Explain how microglia renew themselves. Is it only through proliferation or there is a niche. What is the contribution of bone marrow cells to this?
Ad. According to lineage tracing, parabiosis studies and analyses of repopulation of microglia after CSF1R inhibition, microglia renew themselves by proliferation from existing microglia ( doi.org/10.1016/j.neuron.2014.02.040). The current understanding is that the contribution of bone marrow cells is negligent (doi.org/10.1038/s41593-018-0123-3). Only a pool of BAMs is repopulated during lifetime to some extent via the choroid plexus. In some pathologies (AD models) microglia proliferation and migration to the affected areas was noted. In gliomas, CD11b+ consistently show proliferative gene expression signature. the expression of nestin, which is restricted to populations of neural stem or progenitor cells a corroborating evidence that microglial progenitors were not hematopoietic in nature. L66-69. To the relevant fragment we added: “The expression of a progenitor marker, nestin, is a corroborating evidence that microglial progenitors were not hematopoietic in nature”.
To our knowledge, the effects of a niche have not been studied, so it is hard to speculate on its influence. A recent study addressed this issue but the results suggesting that in Cx3cr1CreER/+R26DTA/+ mice an empty niche allows circulating monocytes to infiltrate the brain, adapt microglia-like signature and give rise to long-lived microglia-like cells (doi.org/10.1038/s41467-018-07295-7). Studies of a role of the niche derived TGF-β signaling in CX3CR1+ monocyte-derived macrophages were performed under pathological condition. Altogether, both lineage tracing and parabiosis studies, and a study by Bennett et al. who transplanted microglia or peripheral immune cells into the empty microglial niche in Csf1r–/– mice suggest a lack of the significant contribution from BM-derived monocytes under physiology, therefore it is an interpretation we present in the manuscript.
- Line 61: It may be worth mentioning a study that used busulfan BMtransplanted to trace monocytes and differentiate microglia. In the absence of pathology, monocytes from the blood did not participate in microglia population after depletion. However, this was not the case in pathology PMID: 30442669.
Ad. We thank the reviewer for indication to discuss microglia and monocyte infiltration into the retina and retinal pathologies. We introduced an appropriate information to the revised version: “Microglia are present at the retina, express the CX3C chemokine receptor 1 (CX3CR1) and undergo morphological activation after corneal injury. Fate-mapping using CX3CR1+/EGFP::CCR2+/RFP reporter mice and BM chimeras confirmed that peripheral monocytes/macrophages do not enter into retina under physiological conditions. When busulfan induced myelodepletion was followed by BM transplantation peripheral CCR2+ CX3CR1+ monocytes migrated to the optic nerve but not to the retina under steady-state conditions. Ocular injury led to population of the retina by peripheral CCR2+ CX3CR1+ monocytes that differentiated to microglia-like CCR2− CX3CR1+ cells. Increased monocyte/macrophage trafficking causes microglia activation and elevation of inflammation (10.1016/j.ajpath.2018.03.005). After depletion of microglia with CSF1R inhibitor (PLX5622), even in the absence of ocular injury, peripheral monocytes repopulated the retina. After ocular injury, the engrafted peripheral monocytes were resistant to CSF1R inhibitor and retained a proinflammatory phenotype, expressing high levels of MHC-II, IL-1β, and TNF-α twenty weeks after the injury (10.1073/pnas.1807123115).
- Line 116: None of the fate mapping techniques are perfect and the possibility that peripheral monocytes may still infiltrate in amyloid deposits.
Ad. We agree with the reviewer as to the limitations of specific approaches and we state in the manuscript L173: “A major challenge in the functional analysis of microglia in diseases is the lack of good experimental systems that allow to discriminate between microglia and monocyte-derived macrophages. While depletion of microglia or macrophages with CD11b-based approaches and other myeloid marker genes provided some interesting clues, these models lack microglial specificity, and target other CNS and peripheral cell types”.
Therefore, in the discussed fragment we specifically mentioned how the authors achieved their conclusions. This was in vivo single-cell imaging in triple-transgenic CD11b-CreERT2;R26-tdTomato;APPPS1 mice brains and confocal imaging of tdTomato microglia clearly demonstrated their microglia-like morphology. At least to us it was a convincing evidence.
- Line 134: In is worth looking at the following two papers. PMID: 30442669; 30541880
Ad,. Thank you for this suggestion. We added the relevant fragment. Please see our comment 2.
- Line 168. I have the feeling that GAM was not spelled-out before use
Ad. “GAMs” abbreviation is now defined when it appears for the first time.
- Lines 243-254. At least for the retina, this has been challenged. Recent reports suggest that PLX5622 affects also peripheral cells PMID: 32900927. This could have been the reason of delayed monocyte contribution in the repopulation after microglia depletion.
Ad. We agree with this comment. Indeed, PLX5622 affect also non-microglial immune cells. The commentary has been added: “ Nevertheless, as CSF1R inhibition by PLX5622 causes not only microglia depletion, but also reduces the number of peripheral monocytes, hematopoietic progenitor and stem cells, it cannot be excluded that the depletion of BM monocytes affects their contribution to CNS-macrophages repopulation.”
- Lines 568-574. The possibility that all these genes are transiently expressed by the same cells should be discussed. The gene expression signature may not be the most accurate way to associate cell lineage.
Ad. In this fragment we discussed a set of the genes which comes from a single cell study and the expression of those genes has been corroborated on in RNA-seq performed on single nuclei from post-mortem frozen tissue from patients with low and high amyloid burden. Therefore, by definition these genes are expressed by the same cells. As in both studies RNA-seq was performed at a single time point, we can’t predict if those genes are transiently expressed or it represents a permanent signature of those cells. scRNA-seq reveals both a cell identity and a transient functional state, as it is in the case of DAMs.
- Lines 712-727. CSF1R inhibition (genetic and PLX5622) affects non microglia cells PMID: 33203068, 32900927, 33446487. This may confound experimental data.
Ad. Indeed, recent studies showed that PLX5622 may deplete not only microglia abut all other myeloid cells. The same is true for genetic ablation which would affect all myeloid lineage cells, therefore fate tracing studies should be combined with busulfan induced myelodepletion followed by bone marrow transplantation for CNS injury experiments, if the experimental results are to be correctly interpreted. We think that those studies combined with insights from single-cell omics may greatly contribute to understanding the heterogeneity of innate immune infiltrates in neuropathological conditions. Specific depletion of either microglia or monocytes-derived macrophages (MDM) cannot be obtained with CSF1R inhibition, as both microglia and MDMs depend on CSF1R signaling for survival. Populations encompassed by the term “GAMs” are now specified.
Some more papers worth reviewing:
Ad. We thank the reviewer for consideration and helpful suggestions. Some papers are discussed in the revised version. We had to omit a large body of references and rely of some excellent reviews to keep this review concise, and focus on insights from single cell omics.
Reviewer 2 Report
This is a very interesting and timely review, highlighting single cell omics to precisely detect different microglia subpopulation and discriminate them from monocytes/macrophages.
The article is well structured starting from the description of microglia origin, localization during brain development and physiological functions and making a correlation with the expression of specific markers, followed by the analysis of the different subpopulations in CNS diseases, and their possible targeting as therapeutic approach.
In my opinion the article has value and could provide relevant information for a wide audience.
However, the following issues should be addressed before publication:
- The use of a high number on non-classical abbreviations requires an abbreviation list at the beginning of the manuscript
- Similarly, it would be very useful to provide a table with the description of all the different genes mentioned in the paper
- It is not clear what the blue squares in table 1 mean, since no legend is provided for them.
- If available, it would be nice to have some mechanisms/effects of the different populations (as done in the paragraph on AD, about the “positive” role of microglia clones for Abeta clearing). For ex. is it known the molecular basis controlling the different microglia activation in IDH1 wt and mut gliomas?
- The sentence at lines 414-417 need to be reformulated since it is not clear which IDh1 variant (w tot mut) shows reduced monocytes/macrophages
Author Response
Reviewer 2
This is a very interesting and timely review, highlighting single cell omics to precisely detect different microglia subpopulation and discriminate them from monocytes/macrophages.
The article is well structured starting from the description of microglia origin, localization during brain development and physiological functions and making a correlation with the expression of specific markers, followed by the analysis of the different subpopulations in CNS diseases, and their possible targeting as therapeutic approach. In my opinion the article has value and could provide relevant information for a wide audience.
However, the following issues should be addressed before publication:
- The use of a high number on non-classical abbreviations requires an abbreviation list at the beginning of the manuscript
Ad. We greatly appreciate reviewer comments and careful review of this manuscript. Their feedback was very helpful. We addressed all comments and made requested amendments to the manuscript .
We made sure that in the revised the non-classical abbreviations are defined in parentheses the first time they appear in the manuscript, as specified in the author guideline of the journal.
- Similarly, it would be very useful to provide a table with the description of all the different genes mentioned in the paper
Ad. We provided the full names of important genes, they are given in the text while discussed a first time. While we think it’s important to show gene/mRNA found in different studies to make comparison between datasets, it is not informative to discuss names of hundreds of genes/mRNAs. We do not think it is practicable to provide a table with the description of all the different genes mentioned in the paper. We provide a supplementary table 1 with abbreviations for names of genes/proteins that appear repeatedly in the text.
- It is not clear what the blue squares in table 1 mean, since no legend is provided for them.
Ad. Blue squares indicate that a given study included cells from a given condition. The figure legend has been added to the Table 1.
- If available, it would be nice to have some mechanisms/effects of the different populations (as done in the paragraph on AD, about the “positive” role of microglia clones for Abeta clearing). For ex. is it known the molecular basis controlling the different microglia activation in IDH1 wt and mut gliomas?
Ad. We extended a description of the mechanisms controlling immune cell composition and their impacts on glioma microenvironment in IDH1 wt and mut gliomas.
- The sentence at lines 414-417 need to be reformulated since it is not clear which IDh1 variant (w tot mut) shows reduced monocytes/macrophages
Ad. The sentence is rephrased in the revised version.